# Screening of *Lactiplantibacillus plantarum* with High Stress Tolerance and High Esterase Activity and Their Effect on Promoting Protein Metabolism and Flavor Formation in *Suanzhayu*, a Chinese Fermented Fish

**DOI:** 10.3390/foods11131932

**Published:** 2022-06-29

**Authors:** Aoxue Liu, Xu Yan, Hao Shang, Chaofan Ji, Sufang Zhang, Huipeng Liang, Yingxi Chen, Xinping Lin

**Affiliations:** 1National Engineering Research Center of Seafood, Collaborative Innovation Center of Provincial and Ministerial Co-Construction for Seafood Deep Processing, Liaoning Province Collaborative Innovation Center for Marine Food Deep Processing, School of Food Science and Technology, Dalian Polytechnic University, Dalian 116034, China; liuaox1998@163.com (A.L.); yanxu8088@gmail.com (X.Y.); sh18340804606@163.com (H.S.); jicf@dlpu.edu.cn (C.J.); zhangsf@dlpu.edu.cn (S.Z.); lianghp@dlpu.edu.cn (H.L.); chenyx@dlpu.edu.cn (Y.C.); 2Department of Agricultural, Forest, and Food Science, University of Turin, Grugliasco, 10095 Turin, Italy

**Keywords:** *Lactiplantibacillus plantarum*, fermented fish, esters, volatile compounds, starter culture, protein metabolism

## Abstract

In this study, three *Lactiplantibacillus plantarum*, namely 3-14-LJ, M22, and MB1, with high acetate esterase activity, acid, salt, and high-temperature tolerance were selected from 708 strains isolated from fermented food. Then, *L. plantarum* strains MB1, M22, and 3-14-LJ were inoculated at 10^7^ CFU/mL in the model and 10^7^ CFU/g in actual *Suanzhayu* systems, and the effects during fermentation on the physicochemical properties, amino acid, and volatile substance were investigated. The results showed that the inoculated group had a faster pH decrease, lower protein content, higher TCA-soluble peptides, and total amino acid contents than the control group in both systems (*p* < 0.05). Inoculation was also found to increase the production of volatile compounds, particularly esters, improve the sour taste, and decrease the bitterness of the product (*p* < 0.05). *L. plantarum* M22 was more effective than the other two strains in stimulating the production of isoamyl acetate, ethyl hexanoate, and ethyl octanoate. However, differences were discovered between the strains as well as between the model and the actual systems. Overall, the isolated strains, particularly *L. plantarum* M22, have good fermentation characteristics and have the potential to become excellent *Suanzhayu* fermenters in the future.

## 1. Introduction

Fermented foods are the product of the combined action of microorganisms and endogenous enzymes, and their flavor and nutrient content differ significantly from the raw materials. It has been discovered that large-scale screening of dominant strains and adding them to fermented foods can promote flavor formation, change the texture and nutrient content of food, and effectively inhibit the growth of harmful bacteria and spoilage bacteria, thereby extending food shelf life. For example, Ho et al. selected a large number of *Lactococcus lactis* and found wild *Lc. Lactis* strains 417 and 537 were able to inhibit the proliferation of *Listeria monocytogenes* in cheese [1]. Zeng et al. isolated 52 yeast strains from *Suanyu* and tested their temperature, pH, and salinity. They identified dominant microorganisms with salt, acid, and thermal tolerance that can survive and function in the *Suanyu* fermentation system, providing strain assurance for future production and fermentation [2]. Thin inoculated a soy sauce model system with two salt-tolerant yeasts isolated from soy sauce moromi samples. It was discovered that inoculation increased the production of volatile flavor compounds such as ethanol, alcohols, furanone, esters, aldehyde, acid, pyrone, and phenols, which are important characteristic flavor compounds in soy sauce [3]. Inoculating Moschofilero wines with the thermotolerant lactic acid bacteria *Lachancea thermotoleransin* SQL increased concentrations of damascenone and geraniol (the main terpenes of Moschofilero), which provide typical floral and fruity aromas to Moschofilero wines, according to Aspasia Nisiotou [4]. According to Visessanguan et al., inoculation of *Lactobacillus campestris* in pork sausage, Nham, significantly reduced the pH, increased the lactic acid content, and prompted protein degradation rate and product flavor formation [5].

Esters play an important role in fermented foods, such as fermented fish [6], beer [7], dairy products [8], wine [9], etc. Esters are primarily formed by microorganisms or endogenous enzymes. Esterase is a hydrolase that can hydrolyze some water-soluble esters, but it can also catalyze the synthesis of esters under certain conditions. Furthermore, in other research, microbial esterase was found to play an important role in the synthesis of ester compounds in fermented foods [10,11,12]. In recent years, lactic acid bacteria, as GRAS strains (Generally Recognized as Safe), have attracted much attention. The addition of lactic acid bacteria containing esterase as a starter culture can enhance the flavor of the product by producing more esters at a faster fermentation rate. In milk Swiss-type cheeses [13], *Lactobacillus paracasei* Lc45 and *Lactobacillus rhamnosus* Lr46 promoted the formation of esters, such as ethyl octanoate and ethyl acetate. In wine [14], esterase in a group inoculated with *Lactobacillus hilgardii* Lac34 have the potential to reduce short-chain ethyl esters such as ethyl acetate. Yang et al. also found that the content of esters, especially ethyl lactate, ethyl acetate, and isoamyl acetate, increased significantly when sauerkraut was inoculated with *L. plantarum* [15].

*Suanzhayu* is a popular fermented fish from southwestern China that has a distinct sour flavor and a long shelf life. *Suanzhayu* is a dish made with freshly caught grass carp. The carp were de-headed, de-tailed, and gutted, then cut into pieces and mixed with rice flour, salt, and spices (methods vary by region) and layered into a filled jar for fermentation [2,16,17,18]. Natural fermentation, on the other hand, is now primarily used, which has a long fermentation cycle, inconsistent quality between batches, and insufficient flavor production. *Suanzhayu*’s current research focuses on the relationships between physicochemical property changes, microbial structure evolution, and volatile flavor compounds during fermentation [17,19,20,21]. *Suanzhayu* starter cultures, particularly those that promote the synthesis of ester compounds, are still uncommon. Gao et al. investigated the ester synthesis ability of fermenters (*Lactobacillus plantarum* 120, *Staphylococcus xylosus* 135, and *Saccharomyces cerevisiae* 31) for the first time in the study of *Suanyu*’s starter culture [22]. The ability of various strains to synthesize esters was primarily investigated using buffer systems containing acyl donors (acetic acid/glyceryl triacetate/acetyl-CoA) and aliphatic alcohols (C2–C6). More research is needed, however, because the buffer system differs from that of fish, and the effect of ester synthesis under the influence of other microorganisms has not been taken into account. The goal of this study was to find ester-producing lactic acid bacteria (LAB) with good fermentation abilities. To begin, we tested 708 LAB strains isolated from fermented foods for acetate esterase activity, salt tolerance, acid tolerance, and high-temperature tolerance in our laboratory. Three *L. plantarum* strains, namely MB1, M22, and 3-14-LJ, were chosen for their excellent fermentation properties. The effect of the three stains on the quality and flavor formation of *Suanzhayu* was investigated using a model and a real-world system to develop a superior starter culture for future *Suanzhayu* production.

## 2. Material and Methods

### 2.1. Large-Scale Screening of Strains with High Salt, High Acid, and High-Temperature Tolerance and High Esterase Activity

For large-scale screening, 708 strains isolated from Chinese fermented meat products (sour meat) and fermented fish products (*Suanzhayu* and *Chouguiyu*) stored in our lab were used. Strains were first cultured for 16 h in 96-well plates in their original isolated culture media, such as MRS, PCA, or LB [14]. The strains were collected by centrifugation (5000× *g* rpm, 5 min, 4 °C) and suspended in 0.9% (*w*/*v*) NaCl, and diluted to a final *OD*_600nm_ of 0.5. The diluted culture was then inoculated at 5% (*v*/*v*) into a 1.0 mL LB medium. The original culture conditions are 1% NaCl, pH 7.0, and 37 °C cultivation. For the test of high salt, high acid, or high-temperature tolerance, the corresponding condition was changed into 8% NaCl, pH value at 4.0, or cultivated at 42 °C, respectively, with the other conditions staying the same. After 24 hours of cultivation, the absorbance at *OD*_600nm_ was recorded. For the determination of esterase activity, 100 μL diluted broth was mixed with 860 μL of 0.1 M Mcllvane buffer (0.1 M citric acid, 0.2 M K_2_HPO_4_, pH = 5.0) and 40 μL of *p*-nitrophenyl acetate (Aladdin, Shanghai, China) stock solution (25 mM). After that, this reaction was held at 37 °C for 2 h, and the absorbance at *OD*_350nm_ was recorded [23]. All the above centrifugations, dilutions, inoculations, and assays were performed on Biomek i7 (Beckman, California, America).

Venn diagrams were used to analyze the top 80 strains with high salt, high acid, and high-temperature tolerance and high esterase activity. For the next stage of the study, 15 strains with high esterase activity and at least two levels of stress tolerance were chosen (Table 1).

### 2.2. Growth Characteristics of Strains under Different Conditions and Re-Measurement of Esterase Activity

In 200 μL of LB medium, 15 strains were inoculated with a 2% (*v*/*v*) inoculum. The medium was set at different NaCl concentrations (8%, 6%, 4%) (*w*/*v*), pH values (7.0, 4.0, 3.0), and cultivation temperatures (20 °C, 37 °C, 42 °C) to determine the strains’ growth in response to different stresses. The absorbance at *OD*_600nm_ was measured using BioScreen C MBR (Oy Growth Curves Ab Ltd., Turku, Finland) in the experiments described above. For a total of 48 h, all strains were incubated to the plateau stage. The esterase activity was measured again in triplicate, as described above. The dry cell weight was determined using a 20 mL culture that had been dried to a constant weight. One unit of enzymatic activity was defined as *p*-nitrophenol (μL) released per minute per gram of dry cell weight.

Three strains of *L. plantarum,* namely M22, MB1, and 3-14-LJ, exhibited better resistance to stress. Among them, *L. plantarum* M22 and 3-14-LJ showed significantly higher esterase activity than MB1. These three strains were used for further studies.

### 2.3. Model System and Actual System of Suanzhayu

Wang’s method [24] was used to create the model system with some modifications. Fresh grass carp fish meat was gutted, boned, and chopped and then mixed with an equal weight of 3% sodium chloride solution, filtered, and centrifuged to obtain the liquid supernatant before autoclaving at 120 °C for 30 min. Groups M1, M2, and M3 were formed by inoculating *L. plantarum* strains MB1, M22, and 3-14-LJ at 10^7^ CFU/mL, respectively. The MC group consisted of the above liquid supernatant that had not been inoculated. These groups were fermented for 16 days at 25 °C in sealed tubes. Inoculation and fermentation were absent in the MF group. Fresh grass carp was gutted, boned, and cut into 2.5 cm^3^ pieces for *Suanzhayu*’s actual system. Then, 200 g of fish were mixed with 3% salt (*w*/*w*) and 30% rice flour (*w*/*w*). *L. plantarum* strains MB1, M22, and 3-14-LJ were inoculated at 10^7^ CFU/g and designated as groups A1, A2, and A3, respectively. The AC group was naturally fermented. In sealed jars, these groups fermented for 16 days at 25 °C. The AF group lacked inoculation and fermentation.

### 2.4. Determination of pH Values, Protein Contents, and TCA-Soluble Peptides Contents

Samples (2 g) were homogenized with 20 mL water for pH determination with a pH meter (Five Easy Plus FE28, Mettler Toledo, Greifensee, Switzerland). For protein content analysis, 2 g samples were homogenized with 20 mL of PBS and centrifuged at 12,000× *g* for 6 min and determined with the Bradford method. Samples (2 g) were homogenized with 8 mL of 10% TCA (*w*/*v*) and centrifuged at 2000× *g* for 10 min. TCA-soluble peptides were determined with the Folin-phenol method according to Yang’s [25] and expressed as μmol Tyr/g.

### 2.5. Determination of the Free Amino Acid Concentrations

The amino acids were determined using Yang’s method [25]. First, 3 g samples were homogenized for 10 min in 12 mL of deionized water and acetone (Sinopharm Chemical Regent) before being evaporated to dryness in a 60 °C water bath. After that, samples were redissolved with acetone and filtered by a 0.22 μm organic filter membrane (Tianjin Branch Billion Lung Co., Ltd., Tianjin, China) and measured with an amino-acid analyzer (LA8080, Hitachi Ltd., Tokyo, Japan). The FFA was expressed as mg/100 g.

### 2.6. Volatile Compounds Analysis with GCMS

For GCMS analysis, 2 g samples were mixed with 10 μL of 100 mg/L iso-octyl alcohol as the internal standard. Divinylbenzene/carboxen/polydimethylsiloxane (DVB/CAR/PDMS Supelco Inc., Bellefonte, PA, USA) fiber was used to extract volatile compounds at 60 °C for 70 min. The capillary column was HP-5MS (30 m × 250 μm × 0.25 μm, Agilent Technologies Inc., OH, USA). The initial temperature was 30 °C for 5 min, then up to 50 °C at 3 °C/min for 3 min, then to 5 °C to 150 °C, and then to 250 °C at 20 °C for 5 min. The identification of volatile organic compounds (VOCs) was based on mass spectroscopy information in the NIST11 library (Agilent Technologies, Inc., Santa Clara, CA, USA) in combination with the retention index (RI). The RI value was calculated using C7-C30 saturated alkanes (Sigma-Aldrich, Sigma-Aldrich, Inc., St. Louis, MO, USA) under the same GC conditions using Cheok’s method equation [26]. The qualitative analysis of esters (isoamyl acetate, ethyl caproate, and ethyl hexanoate) was carried out with standards (Sigma-Aldrich, Sigma-Aldrich, Inc., St. Louis, MO, USA). To quantify the VOCs with internal standards, a semi-quantitative analysis method was used. Odor activity value (OAV) was calculated using the formula OAV = C/OT, where C represented VOC concentration, and OT represented its odor threshold in the public literature [27].

### 2.7. Electronic Tongue Measurement

Electronic tongue measurement was according to the method of Yang [25]. Samples (30 g) were homogenized with 100 mL of water and then filtered for determination with TS-5000Z (Insent, Kanagawa, Japan).

### 2.8. Data Analysis

Significance analysis, variance analysis, and correlation analysis were carried out using the Statistical Package for the Social Sciences (SPSS 20.0, IBM Corp., Chicago, IL, USA). The histogram and radar map were prepared using Origin 9.1 (OriginLab Corp., Northampton, MA, USA). The heat map and cluster analysis were performed by TBtools (version 0.6652). The Venn diagram was created at: https://bioinfogp.cnb.csic.es (accessed on 4 June 2021). The values are shown as mean ± standard deviation (SD), and significant differences in mean values were determined using one-way analysis of variance (ANOVA) and Duncan’s multiple range test. Results were considered as significant when *p* < 0.05. All the experiments were carried out in triplicate.

## 3. Results

### 3.1. Selection of Strains with Salt, Acid, and Temperature Toleranc, and High Esterase Activity

Figure 1A depicts the results of the primary selection from 708 strains. Salt, acid, and temperature tolerance and high esterase activity were found in strains of MC4, MB1, MC2, and MH4 (Figure 1A-area A); acid and temperature tolerance and high esterase activity were found in strains of M51, M39, M53, and M22 (Figure 1A–area B); strains of 3-14-LJ, 2-5-LJ, 3-19-LJ, 18-1-LJ, 1-26-LJ, 14-1-LJ, and M50 were characterized with temperature tolerance and high esterase activity (Figure 1A–area D). Among these 15 strains, except for one *L. pentosus* (3-14-LJ) and one *L.*
*sake* (14-1-LJ)*,* the rest are all *L. plantarum*.

Through secondary selection, three out of fifteen strains, namely *L. plantarum* M22, *L. plantarum* MB1, and *L. plantarum* 3-14-LJ, were chosen for their excellent growth ability in culture media with pH of 4.0 and 6.0, NaCl concentrations of 4% and 6%, and temperatures ranging from 20 °C to 42 °C. The strains, however, cannot grow well at pH = 3 or 8% NaCl. Figure 1B shows the esterase activity of 15 strains ranging from 1.66 to 10.06 U (M/g/min/CDW). Among the fifteen strains, the esterase activity of 3-14-LJ (10.06 U) was significantly higher than that of others, followed by M22 (8.62 U). And MB1 was the smallest (1.66 U) (*p* < 0.05).

### 3.2. Changes in pH, Protein Contents, and TCA-Soluble Peptides

The pH of the inoculated groups (M1, M2, M3) dropped from 6.62 to around 4 (3.64–4.06) in the model system (Table 1), while the pH of the naturally fermented group (MC) was 5.13 at the end of the fermentation. There was a significant difference between the groups with and without inoculation (*p* < 0.05). Similarly, in the actual system of *Suanzhayu* (Table 1), the pH dropped significantly to 3.67–4.02, with the lowest found in group A3, which was inoculated with 3-14-LJ. The results show that inoculating *Lactobacillus* could lower the pH of the samples in either the model or the realistic system, which could be due to the strong ability of acid production.

In the model system (Table 1), after fermentation, the protein content of inoculated groups (M1, M2, M3) dropped from 38.80 mg/kg to around 1 mg/kg (0.75–1.91), while in the naturally fermented group (MC), it was 25.14 mg/ kg. The significance between the groups with and without inoculation (*p* < 0.05) suggests that the inoculation of *Lactobacillus* could promote the degradation of proteins. The TCA-soluble peptides of M2 and MC increased significantly after fermentation, with M2 having the highest level. Similarly, in the actual *Suanzhayu* system (Table 1), protein content decreased, while TCA-soluble peptides increased when compared to fresh *Suanzhayu*. The A2 had the least protein and the most TCA-soluble peptides. The TCA-soluble peptides of M1, M3, and A1, A3 were lower than in the control groups, which could be attributed to strain variability. Overall, these findings suggest that inoculating *L. plantarum* M22 strains (groups M2 and A2) can promote protein hydrolysis and increase the content of soluble polypeptides.

### 3.3. Changes in Amino Acid Content

Table 1 shows an increase in amino acid content in both the *Suanzhayu* model and the actual system. The amino acid content of M1 and M3 in the model system was significantly higher than that of the control group (*p* < 0.05), whereas the content of M2 (*L. plantarum* M22) was lower (*p* < 0.05). Similarly, the A2 group had the lowest amino acid content in the actual system. In the samples inoculated with *L. plantarum* M22, more proteins were converted into TCA soluble peptides than amino acids, according to the results of protein content and TCA soluble peptides.

In the model system, nine amino acids were detected (Table 2), and the content of bitter amino acids (BAAs) increased, while the content of umami amino acids (UAAs) decreased. In the actual system, 17 amino acids were detected (Table 2). It was discovered that the amino acid content of *Suanzhayu* increased significantly after fermentation. Furthermore, the inoculated group’s UAA content was significantly higher than the control group. At the end of fermentation, BAAs were lowest in the A2 group, indicating that inoculation of *L. plantarum* M22 can reduce BAAs in the actual *Suanzhayu* system, which may be beneficial to the overall flavor.

### 3.4. Changes in Volatile Compounds by GCMS

The model system detected 31 VOCs; the clustering results (Figure 2A) revealed that the samples with inoculation (M1, M2, M3) clustered into one category, while the control sample (MC) and fresh sample (MF) clustered into another. This result indicates that the VOCs in the *Suanzhayu* model system with inoculation differed from the control and fresh systems. In all, 43 volatile compounds were detected in the actual system (Figure 2B); including 10 alcohols, 4 alkanes, 17 esters, 4 aldehydes, 2 ketones, 2 acids, and 6 other flavor substances. The AC and A1 samples clustered together, whereas the A2 and A3 samples contained significantly more VOCs, indicating that inoculation with *L. plantarum* M22 and *L. plantarum* 3-14-LJ could improve the overall compounds in *Suanzhayu*. Appendix A showed that in both the model and actual systems, VOCs in the inoculated groups were higher than in the control group, with the most significant increase in alcohol content. At the same time, inoculation of *L. plantarum* M22 and 3-14-LJ can significantly increase some important compounds that can provide special flavor to *Suanzhayu*, such as octanal, hexanal, 2-pentylfuran, 1-octen-3-ol, and 3-hexannone. When the VOC content of the three strains was compared, the total ester content and total VOC content of the group inoculated with *Lactobacillus plantarum* M22 were the highest in both the model and actual system. This indicated that inoculation could significantly increase the content of VOCs in *Suanzhayu*, with *L. plantarum* M22 having the greatest effect.

It was found that the content and type of esters produced in the model system were significantly less than that in the actual system (Table 1, Appendix A). After fermentation in the model system, the total ester content was significantly higher in the groups with inoculation than in the groups without, with M2 exhibiting the highest total ester content. The ester content in A2 was higher than AC in the actual system, while it was lower in A1 and A3. This demonstrated that LAB, particularly *L. plantarum* M22, can effectively boost ester production in both the model and actual systems. In the model system (Figure 2C), the groups with inoculation produced significantly more isoamyl acetate, ethyl caproate, and ethyl hexanoate than the control group, with M2 having the highest content. In the actual system (Figure 2D), all three esters were detected in the inoculated groups, but no ethyl octanoate was found in the AC group. The A2 group contained the most isoamyl acetate and ethyl hexanoate, followed by AC. Inoculation with LAB, particularly *L. plantarum* M22, was more favorable for the formation of ester compounds when combined.

### 3.5. Results of Electronic Tongue Measurement

As shown in Figure 3, when compared to the control group, the inoculated groups had a significant increase in sourness and a significant decrease in bitterness. The M2 group had the lowest pH in the model system, but it did not have the highest sourness, which could be because of the fact that *L. plantarum* M22 produced some non-sour acidic substances during the fermentation process, such as palmitic acid [28]. Although the BAA content increased significantly after inoculation in both the model and actual systems, the electronic tongue results showed that bitterness decreased in the inoculated groups. By combining the results of both systems, Inoculation was able to improve the acidity and reduce the bitterness of the *Suanzhayu* product.

## 4. Discussion

Three strains of *L. plantarum*, namely MB1, M22, and 3-14-LJ, were chosen from 708 and inoculated in the *Suanzhayu* model and actual systems, respectively. Inoculation was found to accelerate the rate of pH decline in the system, promote protein degradation, and increase the content of TCA soluble peptides, amino acids, and VOCs, particularly esters, in *Suanzhayu*. Furthermore, electronic tongue results revealed that inoculation could increase the acidity and decrease the bitterness of the product. Inoculation helps maintain the stability of the product and improves quality such as taste. Low pH is crucial for the inhibition of spoilage and pathogenic bacteria [29]. A similar phenomenon that inoculation of LAB could reduce the product’s pH, allowing rapid acidification of the product, was also reported in Zeng’s study [30]. Furthermore, it was demonstrated that decreasing the pH aids protein hydrolysis and increases the content of TCA-soluble peptides. *Suanyu* (a traditional Chinese low-salt fermented fish) proteins gradually hydrolyzed, and TCA-soluble peptides and FAAs content increased with decreasing pH during fermentation. The protein concentration was lowest at the end of fermentation, while the peptide content was highest [24]. The same phenomena were observed not only in our study but also in many fermentation products. The increase of TCA-soluble peptides and FAAs content play important roles in taste perception. For example, glutamic acid is one of the most important substances for “freshness” [31], while alanine is the main contributor to sweetness [32]. Chen’s study discovered that after LAB inoculation, the content of fresh (Asp, Glu) and sweet (Gly, Ser, Pro, etc.) amino acids increased [33]. The actual *Suanzhayu* system also showed an increase in SAA and UAA. However, the model system lacked Asp, Ser, and Glu. This might be a result of the single-strain inoculation, which might not accurately reflect the product’s true specialty. It has been observed that under low pH stress, LAB could utilize Glu to create gamma-aminobutyric acid (GABA) to alleviate the stress [34]. This could also be because the model system’s pH (3.64) is lower than the actual system’s pH (4.02). Immunization greatly improved the product’s sourness while also significantly lessening its bitterness. This might be a result of the acid that the LAB’s carbohydrate metabolism produces, which has been observed in numerous fermented goods [35]. The decrease in bitterness in the inoculated group may be because bitterness is determined not only by BAAs’ content but also by some bitter peptides, etc. [14,36]. The lower ability of degradation of bitter peptides or other bitter substances without LAB may result in a higher bitter taste.

Inoculation had a significant effect on the production of esters. In the model system, isoamyl acetate, ethyl caproate, and ethyl caprylate were not detected in either the fresh or the control group, while all three inoculated groups produced higher amounts of these three esters. Similar results were obtained by Gao et al. after inoculation of LAB in the *Suanyu* model system. Inoculation with *L. plantarum* M22 was most effective in producing these esters compared with the other two strains [37]. Esters with short-chain acids are represented with a fruity aroma, while esters containing long-chain acids give a fatty odor [38]. Alcohols were also enhanced with inoculation, especially for 3-methyl-1-butanol, 1-hexanol, 1-octen-3-ol, and phenyl ethanol [39]. 1-octen-3-ol, commonly found in aquatic products, has a low odor threshold and can provide mushroom and fruit aromas [40]. Aldehydes are also influenced by inoculation, such as significantly increased glutaraldehyde, hexanal, heptanal, octanal, nonanal, and (E)-2-octanal, all derived from the oxidation of unsaturated fatty acids [41]. Hexanal can provide a distinctive fatty, fishy, and nutty taste to the products [42]. Other substances, such as 3-hexanone and 3-octanone, help to increase products’ flavor of butter, cream, and fruits [43]; 2-pentylfuran [42] provides the aroma of legumes and fruits. Inoculation elevated the content of these volatile compounds and improved the flavor of *Suanzhayu*.

LAB strain differences have varying effects on product quality. The amount of exogenous and endogenous enzymes in each strain differed significantly. The M22 group of *L. plantarum* had the highest soluble peptide content and the lowest amino acid content, whereas the 3-14-LJ and MB1 groups had lower TCA-soluble peptide content and higher amino acid content. This could be because *L. plantarum* M22 has a lower peptidase activity or a higher rate of amino acid degradation than the other two strains. Maria’s study discovered the same differences [44]. They discovered that different strains of *Penicillium* may have different enzyme systems, resulting in a different degree of protein degradation, peptide, and amino acid content of the product when they studied the effect of 14 strains of *Penicillium* on the quality of fermented sausages. The effect of strains on VOC formation varies as well. In both the model and actual systems, the *L. plantarum* M22 group had the highest total ester content compared to the other two inoculated groups. It is thought that the three starter cultures had different esterase activities. Pang et al. inoculated 36 *L. planturum* for Chinese liquid production and discovered that the content of ethyl acetate increased significantly across all inoculation groups. The group that received *L. planturum* Lp1 inoculation had the most significant results that did exist among strains, such as that the esterase activities of *L. planturum* Lp1 was 73 U/mL higher than that of *L. planturum* Lp2 [45]. Procopio et al. also found that ester production showed strain specificity in beverages [46].

It is normal for the results of the actual and model systems to differ slightly. The ester content and species in the model system were found to be significantly lower than those in the actual system. Furthermore, the actual system had higher total contents of the three esters—isoamyl acetate, ethyl hexanoate, and ethyl octanoate—than the model system. Inconsistencies between actual and model systems were also discovered in other studies. Liliana discovered, for example, that if the model fermented sausages were inoculated with *Debaryomyces hansenii*, they contained fewer volatile compounds than the actual system. They hypothesized that it was caused by the interaction of other microorganisms with *D. hansenii* [47]. The model system only has one inhabitant, whereas the actual system has a mixed niche, producing an incomplete representation of the ecosystem. Furthermore, taste creation is a combination of the outcomes of the metabolic network, and the inoculated LAB may have intricate interactions with naturally existent residents in the actual system. Because of this, simple model systems may be applied for convenience, but their rudimentary structure will not account for strain interactions. Investigating a combination of model and actual systems might be a better course of action.

## 5. Conclusions

In summary, 708 strains were isolated from *Suanzhayu* and sour meat, and finally, *L. plantarum* 3-14-LJ, M22, and MB1, with higher acetate esterase activity, were selected. The three strains were inoculated into the *Suanzhayu* in both the model and actual systems. Based on the findings, it was concluded that LAB can reduce the pH of *Suanzhayu*, promote protein degradation, increase the content of TCA soluble peptides, and elevate the amino acid composition. Inoculation also improved the formation of volatile substances in *Suanzhayu* as well as the modification of flavor substances, particularly esters. LAB can also be used to reduce the bitterness of *Suanzhayu* as well as increase its sourness. The three strains of *L. plantarum* obtained through large-scale screening, especially *L. plantarum* M22, may be good fermenting agents for fermenting *Suanzhayu*, and further consideration can be given to making these agents into direct-injection fermenting agents for commercial production in the future.

## Figures and Tables

**Figure 1 foods-11-01932-f001:**
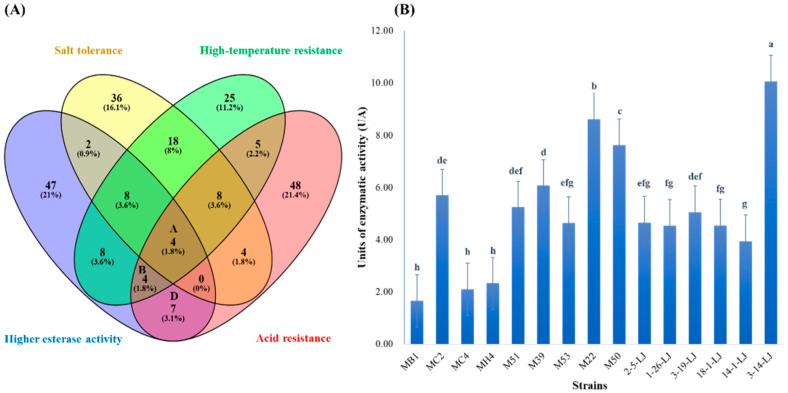
Strain screening result. (**A**) Venn diagram of the top 80 high-temperature-resistant strains, the top 80 salt-tolerant strains, the top 80 acid-resistant strains, and the top 80 strains of esterase activity. (**B**) Results for esterase activity against p-nitrophenyl acetate in the selected 15 strains. Letters “a–h” indicate the significant difference. (*p* < 0.05).

**Figure 2 foods-11-01932-f002:**
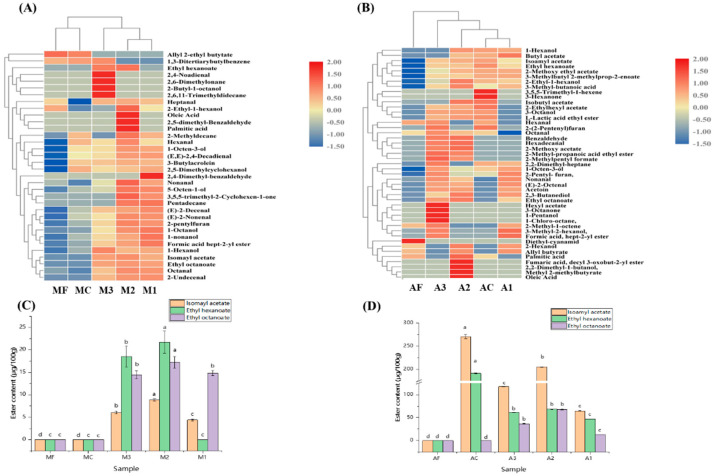
Hierarchical clustering of volatile compounds in model system (**A**) and actual system (**B**) of *Suanzhayu.* Isoamyl acetate, ethyl hexanoate, and ethyl octanoate contents in model system (**C**) and actual system (**D**) of *Suanzhayu*. Letters “a–d” indicate the significant difference (*p* < 0.05). For abbreviations, see Table 1.

**Figure 3 foods-11-01932-f003:**
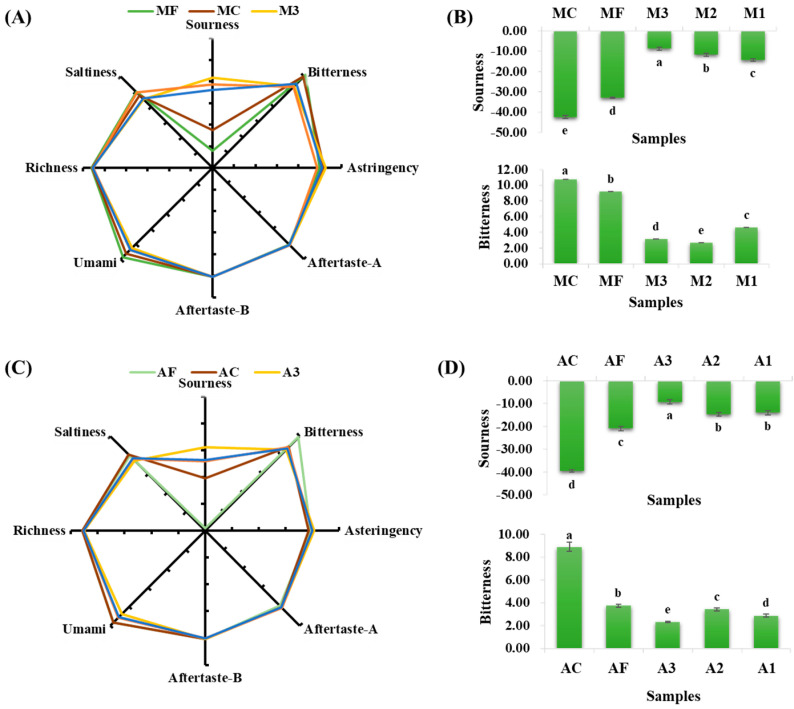
E-tongue analysis of samples from the model system (**A**) and actual system (**C**). The results of the sour characteristics and the bitterness characteristics of E-tongue in the model and actual system are represented in (**B**,**D**), respectively. Letters “a–e” indicate the significant difference (*p* < 0.05). For abbreviations, see Table 1.

**Table 1 foods-11-01932-t001:** Results of pH values, protein contents, TCA-soluble peptide contents, amino acid contents, and total ester contents in the model system and actual system of *Suanzhayu*.

Attribute	Samples									
	Model System of *Suanzhayu*					Actual System of *Suanzhayu*				
	MF	MC	M3	M2	M1	AF	AC	A3	A2	A1
pH	6.62 ± 0.08 ^a^	5.13 ± 0.23 ^b^	3.94 ± 0.02 ^c^	3.64 ± 0.05 ^d^	4.06 ± 0.03 ^c^	6.50 ± 0.08 ^a^	4.07 ± 0.03 ^b^	3.67 ± 0.03 ^c^	4.02 ± 0.08 ^b^	4.02 ± 0.04 ^b^
Protein contents (mg/kg)	38.80 ± 8.11 ^a^	25.14 ± 6.86 ^b^	1.08 ± 014 ^c^	1.91 ± 0.32 ^c^	0.75 ± 0.36 ^c^	400.96 ± 17.57 ^a^	20.5 ± 3.98 ^b^	18.09 ± 4.77 ^b,c^	6.43 ± 2.07 ^d^	9.16 ± 2.19 ^c,d^
TCA-soluble peptide contents (mg/kg)	98.62 ± 3.82 ^c^	126.85 ± 4.80 ^b^	81.73 ± 7.28 ^d^	139.46 ± 5.70 ^a^	81.83 ± 3.53 ^d^	219.92 ± 9.61 ^e^	931.04 ± 36.90 ^b^	752.75 ± 17.35 ^d^	1002.75 ± 29.19 ^a^	840.89 ± 23.92 ^c^
Amino acid contents (mg/100 g)	87.85 ± 4.84 ^c^	95.73 ± 0.34 ^b^	107.07 ± 0.10 ^a^	90.03 ± 4.87 ^c^	102.15 ± 3.62 ^a^	108.75 ± 1.02 ^e^	387.73 ± 3.03 ^c^	411.97 ± 1.02 ^b^	337.28 ± 19.54 ^d^	509.51 ± 1.38 ^a^
Total ester contents (μg/100 g)	22.13 ± 2.60 ^d^	19.05 ± 5.61 ^d^	110.11 ± 10.87 ^b^	150.69 ± 2.07 ^a^	90.36 ± 4.37 ^c^	105.63 ± 31.82 ^c^	2818.33 ± 781.51 ^a,b^	1897.47 ± 776.95 ^b^	3366.07 ± 1090.28 ^a^	1860.21 ± 716.32 ^b^

Letters “a–e” indicate the significant difference in the model system and actual system of *Suanzhayu*, respectively (*p* < 0.05). Abbreviations: M, model system; A, *Suanzhayu*’s actual system; F, fresh meat; C, control, naturally fermented; 3, inoculated with *L. plantarum* 3-14-LJ; 2, inoculated with *L. plantarum* M22; 1, inoculated with *L. plantarum* MB1.

**Table 2 foods-11-01932-t002:** Amino acid composition in the model system and actual system of *Suanzhayu*.

Amino Acids						Samples				
	Model System of *Suanzhayu*					Actual System of *Suanzhayu*				
	MF	MC	M3	M2	M1	AF	AC	A3	A2	A1
Gly	23.67 ± 0.08 ^c^	24.6 ± 0.10 ^b^	25.76 ± 0.02 ^a^	22.05 ± 0.05 ^d^	24.48 ± 0.05 ^b^	9.15 ± 0.01 ^e^	20.47 ± 0.00 ^b^	16.95 ± 0.01 ^c^	10.13 ± 0.07 ^d^	24.31 ± 0.02 ^a^
Pro	ND	ND	ND	ND	ND	1.83 ± 0.00 ^e^	4.83 ± 0.01 ^d^	42.58 ± 0.15 ^b^	35.34 ± 0.01 ^c^	48.57 ± 0.06 ^a^
Ser	1.28 ± 0.01 ^a^	0.26 ± 0.01 ^b^	ND	ND	ND	2.38 ± 0.02 ^b^	2.28 ± 0.05 ^c^	1.98 ± 0.01 ^e^	2.02 ± 0.01 ^d^	3.39 ± 0.00 ^a^
Thr	2.73 ± 0.01 ^a^	1.88 ± 0.00 ^b^	1.27 ± 0.01 ^c^	0.33 ± 0.00 ^e^	1.14 ± 0.01 ^d^	3.42 ± 0.01 ^e^	20.61 ± 0.04 ^b^	19.42 ± 0.03 ^c^	11.98 ± 0.04 ^d^	24.87 ± 0.03 ^a^
Ala	5.78 ± 0.00 ^b^	8.31 ± 0.07 ^a^	4.78 ± 0.01 ^d^	4.58 ± 0.01 ^e^	5.55 ± 0.01 ^c^	16.31 ± 0.04 ^e^	88.59 ± 0.75 ^a^	56.67 ± 0.12 ^c^	49.86 ± 0.38 ^d^	79.62 ± 0.01 ^b^
ΣSAA	33.46 ± 0.10 ^b^	35.04 ± 0.16 ^a^	31.80 ± 0.04 ^c^	26.96 ± 0.04 ^e^	31.17 ± 0.05 ^d^	33.09 ± 0.08 ^e^	136.78 ± 0.85 ^c^	137.60 ± 0.32 ^b^	109.33 ± 0.51 ^d^	180.76 ± 0.12 ^a^
Asp	ND	0.27 ± 0.01 ^a^	ND	ND	ND	0.45 ± 0.00 ^e^	6.47 ± 0.00 ^c^	10.98 ± 0.03 ^b^	5.04 ± 0.00 ^d^	16.11 ± 0.03 ^a^
Glu	0.97 ± 0.03 ^b^	1.41 ± 0.01 ^a^	ND	ND	ND	3.49 ± 0.05 ^a^	7.49 ± 0.17 ^b^	31.47 ± 0.06 ^d^	25.09 ± 0.10 ^c^	36.58 ± 0.07 ^e^
ΣUAA	0.97 ± 0.03 ^b^	1.68 ± 0.02 ^a^	ND	ND	ND	3.93 ± 0.05 ^e^	13.97 ± 0.17 ^d^	42.45 ± 0.09 ^b^	30.14 ± 0.10 ^c^	52.69 ± 0.11 ^a^
Arg	ND	ND	ND	ND	ND	2.78 ± 0.29 ^c^	ND	5.38 ± 0.33 ^b^	16.00 ± 0.24 ^a^	1.88 ± 1.88 ^c^
Met	ND	ND	ND	ND	ND	0.42 ± 0.05 ^e^	27.53 ± 0.06 ^a^	15.93 ± 0.04 ^c^	13.48 ± 0.06 ^d^	23.41 ± 0.02 ^b^
His	48.06 ± 2.63 ^d^	51.24 ± 0.16 ^d^	66.29 ± 0.01 ^a^	57.3 ± 2.47 ^c^	62.16 ± 3.05 ^b^	52.62 ± 0.03 ^a^	9.01 ± 0.05 ^d^	37.46 ± 0.03 ^b^	24.21 ± 2.41 ^c^	22.64 ± 0.30 ^c^
Lle	ND	ND	ND	ND	ND	0.90 ± 0.00 ^e^	24.94 ± 0.01 ^a^	13.18 ± 0.02 ^c^	10.85 ± 0.05 ^d^	22.67 ± 0.22 ^b^
Leu	ND	ND	ND	ND	ND	ND	55.11 ± 0.01 ^b^	44.79 ± 0.11 ^c^	37.97 ± 0.08 ^d^	61.28 ± 0.46 ^a^
Phe	ND	ND	ND	ND	ND	ND	51.88 ± 0.09 ^a^	51.82 ± 0.25 ^a^	45.36 ± 15.53 ^a^	64.40 ± 1.20 ^a^
Val	1.12 ± 0.01 ^b^	1.31 ± 0.01 ^a^	0.78 ± 0.00 ^c^	0.51 ± 0.01 ^e^	0.64 ± 0.00 ^d^	2.49 ± 0.01 ^a^	45.24 ± 0.02 ^e^	22.88 ± 0.04 ^c^	21.50 ± 0.10 ^d^	36.74 ± 0.04 ^b^
Tyr	ND	ND	ND	ND	ND	5.16 ± 1.25 ^d^	5.10 ± 0.02 ^d^	16.89 ± 0.05 ^a^	10.55 ± 0.2 ^c^	12.39 ± 0.54 ^b^
ΣBAA	49.18 ± 2.63 ^d^	52.54 ± 0.16 ^d^	67.07 ± 0.01 ^a^	57.80 ± 2.48 ^c^	62.81 ± 3.05 ^b^	64.37 ± 0.98 ^d^	218.80 ± 0.18 ^b^	208.33 ± 0.85 ^b^	179.94 ± 17.62 ^c^	245.41 ± 0.89 ^a^
Cys	ND	ND	ND	ND	ND	ND	3.67 ± 1.91 ^a^	2.45 ± 0.01 ^a,b^	1.73 ± 0.02 ^b^	3.02 ± 0.05 ^a,b^
Lys	4.24 ± 2.14 ^b^	6.47 ± 0.02 ^a,b^	8.19 ± 0.14 ^a^	5.27 ± 2.35 ^b^	8.18 ± 0.50 ^a^	7.36 ± 0.04 ^d^	14.52 ± 0.06 ^c^	21.16 ± 0.05 ^b^	16.13 ± 2.54 ^c^	27.63 ± 0.26 ^a^
ΣTAA	4.24 ± 2.14 ^b^	6.47 ± 0.02 ^a,b^	8.19 ± 0.14 ^a^	5.27 ± 2.35 ^b^	8.18 ± 0.50 ^a^	7.36 ± 0.04 ^e^	18.18 ± 1.84 ^c^	23.60 ± 0.06 ^b^	17.87 ± 2.52 ^c^	30.65 ± 0.31 ^a^
Total	87.85 ± 4.84 ^c^	95.73 ± 0.34 ^b^	107.07 ± 0.10 ^a^	90.03 ± 4.87 ^b^	102.15 ± 3.62 ^a^	108.75 ± 1.02 ^e^	387.73 ± 3.03 ^c^	411.97 ± 1.02 ^b^	337.28 ± 19.54 ^d^	509.51 ± 1.38 ^a^

Letters “a–e” indicate the significant difference in the model system and *Suanzhayu,* respectively. (*p* < 0.05). ND, not detected; SAA, sweet amino acids; UAA, umami amino acids; BAA, bitter amino acids; TAA, tasteless amino acids. For abbreviations, see Table 1.

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
