# Peer review of "Screening of Lactiplantibacillus plantarum with High Stress Tolerance and High Esterase Activity and Their Effect on Promoting Protein Metabolism and Flavor Formation in Suanzhayu, a Chinese Fermented Fish"

_foods, 2022, doi:10.3390/foods11131932_

Round 1
Reviewer 1 Report
This is with reference to your request for reviewing article with the title “Large-scale screening of Lactiplantibacillus plantarum M22 with high stress tolerance and high esterase activity and promoting protein metabolism and flavor formation in Suanzhayu, a Chinese fermented fish”. I have reviewed the article and submitting my comments and suggestions. The reported work is in the journal’s scope and add to knowledge base. The proposed article provides interesting results and can consider for publication in the journal after minor corrections.
Reviewer’s comments:
· Which statistical model was applied? Kindly mention in methodology section.
· Kindly elaborate the discuss results of Figure 1B. Were all the strains showed significant difference with each other?
· What were the main findings of the study? Kindly write a proper conclusion of the study.
· Kindly add some more recent references.
Author Response
Dear editor and reviewers:
Thank you so much for your email with the reviewers’ comments on our manuscript (ID: foods-1394574). We have studied comments seriously and have made corrections accordingly which we hope to meet with approval. The detailed revision response will be attached in the form of a document.

Reviewer 2 Report
Comments to authors
Journal: Foods
Title: ‘Large-scale screening of Lactiplantibacillus plantarum M22 2 with high stress tolerance and high esterase activity and promoting protein metabolism and flavor formation in Suanzhayu, a Chinese fermented fish’
The current research article focused on identification of ester-producing lactic acid bacteria (LAB) having admirable fermentation properties by testing plenty of LAB strains isolated from fermented foods and three of them exhibited excellent properties regarding quality and flavor formation of Suanzhayu. The research is groundbreaking and interesting, but still, I have follwing questions as mentioned below.
Comments:
1. The title of the article is lengthy, I suggest the authors to make it short and concise.
2. Line 22 ‘following that …’ confusing sentence it should be revise.
3. Line 50-52 ‘Esterases were a type of hydrolytic synthetic…’ rewrite the sentence.
4. Line 68-70 ‘Esterases were a type of hydrolytic synthetic...’ sentence should be revise.
5. The introduction of the manuscript should be improved and well-structured.
6. What is the actual difference between both tables in the paper, both are denoted as table1 Why?
7. From fig. 1 A how can readers know that which strains are acid tolerance or high temperature resistance or other?
8. Figure 2 A, and B are very small difficult to read, I suggest the authors to make it large.
9. Figure 3 should be revised the denotation is confusing figure name should be relevant with the captions.
10. What is the relation between growth and pH?
11. I suggest the authors to mentioned flavor formation results more clearly.
12. The article needs to be revised thoroughly by English experts as it contains many syntax and grammatical errors.
13. Results should be well defined and the discussion part is not closely related to the results, and the focus of the discussion is not focused. So, improve the discussion part according to the current and previous studies.
Author Response

(The authors gave the same response as above.)

Reviewer 3 Report
General comments
The topic is novel, well written, and may contribute to our current understanding of fermented meat products. The text is clear and easy to understand. The manuscript addressed the hypothesis well and concluded with the salient findings.
I recommend its publication after revisions as listed below-
Title: have scope for improvement as Screening of Lactiplantibacillus plantarum M22 with high stress tolerance and high esterase activity and their effect on------- or any other suitable, plz
Also the study also includes three strains, 3-14-LJ, M22 and MB1 and all these three strains are used for fish fermentation.
Abstract:
i. Line 25: mention inoculation rate
ii. Please mention the p value/ level of significance of the results, whether the results are statistically significant.
Keywords:May be improve to more attractive and clear
Introduction
i. Appropriate, but need a description on salt, acid and thermal tolerance
Materials and methods
i. Line 89: which meat products?
ii. Plz give more details of sample size, n and p values and statistical analysis
Results and Discussion
Appropriate and well supported by references
i. Table 1: plz check numbering and also Table 1 given two times
ii. Table S1, S2 and Table 2: I could not find in manuscript file
Summary: may be better stated.
Author Response

(The authors gave the same response as above.)
